# Theophylline Attenuates BLM-Induced Pulmonary Fibrosis by Inhibiting Th17 Differentiation

**DOI:** 10.3390/ijms24021019

**Published:** 2023-01-05

**Authors:** Soo-Jin Park, Hwa-Jeong Hahn, Sei-Ryang Oh, Hyun-Jun Lee

**Affiliations:** Natural Medicine Research Center, Korea Research Institute of Bioscience and Biotechnology (KRIBB), 30 Yeongudanji-ro, Ochang-eup, Cheongwon-gu, Cheongju 28116, Chungbuk, Republic of Korea

**Keywords:** theophylline, pulmonary fibrosis, Th17, TGF-β

## Abstract

Idiopathic pulmonary fibrosis (IPF) is a chronic and refractory interstitial lung disease. Although there are two approved drugs for IPF, they were not able to completely cure the disease. Therefore, the development of new drugs is required for the effective treatment of IPF. In this study, we investigated the effect of theophylline, which has long been used for the treatment of asthma, on pulmonary fibrosis. The administration of theophylline attenuated the fibrotic changes of lung tissues and improved mechanical pulmonary functions in bleomycin (BLM)-induced pulmonary fibrosis. Theophylline treatment suppressed IL-17 production through inhibiting cytokines controlling Th17 differentiation; TGF-β, IL-6, IL-1β, and IL-23. The inhibition of IL-6 and IL-1β by theophylline is mediated by suppressing BLM-induced ROS production and NF-κB activation in epithelial cells. We further demonstrated that theophylline inhibited TGF-β-induced epithelial-to-mesenchymal transition in epithelial cells through suppressing the phosphorylation of Smad2/3 and AKT. The inhibitory effects of theophylline on the phosphorylation of Smad2/3 and AKT were recapitulated in BLM-treated lung tissues. Taken together, these results demonstrated that theophylline prevents pulmonary fibrosis by inhibiting Th17 differentiation and TGF-β signaling.

## 1. Introduction

Idiopathic pulmonary fibrosis (IPF) is a chronic and progressive interstitial lung disease with no effective treatment other than lung transplantation. IPF is caused by the dysregulation of wound healing processes triggered by persistent damage to lung epithelial cells, involving the differentiation and proliferation of fibroblasts, the accumulation of extracellular matrix (ECM), and the recruitment and activation of immune cells, which can lead to fibrotic changes of lung tissues and eventually respiratory failure [1,2]. Although there are several known factors for the initiation and progression of IPF, such as cigarette smoking and lung infections, the etiology and pathogenic mechanisms of IPF remain unclear [3]. Heretofore, two drugs, pirfenidone (PFD) and nintedanib, have been approved for IPF. PFD inhibits collagen production and fibroblast proliferation. Nintedanib is a tyrosine kinase inhibitor and regulates growth factor pathways. Both drugs slow the progression of pulmonary fibrosis and increase survival, but they have side effects and cannot completely cure the disease [4,5]. Therefore, there is an urgent need to develop of a new therapeutic agent for IPF.

One of the major factors involved in pulmonary fibrosis is transforming growth factor-β (TGF-β), and its signaling pathway plays an important role in IPF pathogenesis. TGF-β is produced by most cell types and tissues in fibrotic conditions and induces the activation of epithelial cells and fibroblasts, leading to epithelial-to-mesenchymal transition (EMT), fibroblast-to-myofibroblast transdifferentiation, and collagen and ECM production [6,7,8,9]. Moreover, TGF-β induces the differentiation of Th17 cells. IL-17 is the major cytokine released from Th17 cells [10]. Accumulating evidence indicated that IL-17 is also implicated in the pathogenesis of pulmonary fibrosis [9,11]. In IPF patients, IL-17 was detected in fibrotic foci and was increased in the bronchoalveolar lavage (BAL) fluid [12,13]. IL-17A induced the production of collagen and EMT in epithelial cells in a TGF-β-dependent manner and blocked the IL-17 signaling pathway through the neutralization of IL-17A attenuated BLM-induced inflammation and fibrosis in the lung [10,13]. 

Theophylline is a methylxanthine derivative that has been used to treat asthma for a long time [14]. However, the effect of theophylline on pulmonary fibrosis has not been investigated. Therefore, in this study, the effects of theophylline on pulmonary fibrosis and underlying molecular mechanism were explored using the bleomycin (BLM)-induced pulmonary fibrosis mouse model. 

## 2. Results

### 2.1. Theophylline Improves Lung Function in a BLM-Induced Pulmonary Fibrosis Mouse Model

We first investigated the effects of theophylline on pulmonary fibrosis using a BLM-induced fibrosis mouse model (Figure 1A). The body weight of all experimental groups was reduced by a single intratracheal administration of BLM, then recovered gradually after day 8 (Figure 1B). On the other hand, the relative lung weight was significantly increased by BLM treatment. The oral administration of 20 to 40 mg/kg of theophylline reduced lung weight gain in BLM-treated mice (Figure 1C). Next, lung volumes, compliance, and resistance were measured to analyze the effect of theophylline on BLM-induced lung dysfunction. The mechanical dynamics of the respiratory system in BLM-treated mice were measured by forced oscillation plethysmography Scientific Respiratory Equipment Inc. (SCIREQ). The treatment of BLM significantly reduced lung volume, the distensibility of the respiratory system (Cst), and the compliance of the total respiratory system (Crs). The resistance of the total respiratory system (Rrs), the elastance of the total respiratory system (Ers), the damping of tissue (G), and the elastance of tissue (H) were increased by BLM instillation. These changes were significantly improved after an administration of theophylline in BLM-treated mice. The oral administration of pirfenidone (PFD), a therapeutic drug for IPF, also suppressed BLM-induced lung fibrosis in this experimental model (Figure 1D,E). These data indicated that theophylline improved the impaired mechanical function of the respiratory system in BLM-induced pulmonary fibrosis.

### 2.2. Theophylline Attenuates Pulmonary Fibrosis Progression in BLM-Induced Mice

BLM treatment induces inflammatory cell accumulation, the deposition of extracellular matrix and collagen, and the destruction of lung structures leading to pulmonary fibrosis. To confirm the accumulation of inflammatory cells in the fibrotic lung, the number of total cells and differential cell population was counted in BLAF. The number of total cells, macrophages, and lymphocytes was significantly increased in BLM-treated mice compared to control mice. However, in theophylline- and pirfenidone-treated mice, inflammatory cell infiltration in the lungs was decreased (Figure 2).

To determine the effect of theophylline on the fibrotic lung, histopathological analysis was performed using H&E and Masson’s trichrome stain. BLM treatment induced inflammatory cell infiltration, collagen accumulation, and the destruction of the lung structure in the lung tissue. Theophylline treatment attenuated BLM-induced histological changes (Figure 3A,B). BLM treatment quantitatively elevated the amount of collagen in the lung tissues. Theophylline administration markedly inhibited BLM-increased collagen production (Figure 3C,D). One of the characteristics of pulmonary fibrosis is epithelial-to-mesenchymal transition (EMT), which is the process of losing epithelial proteins and moving to a more mesenchymal phenotype. The typical feature of EMT is the upregulation of N-cadherin followed by the downregulation of E-cadherin. To investigate the effects of theophylline on EMT, the protein levels of N-cadherin and E-cadherin in lung tissue was analyzed by Western blot. The instillation of BLM upregulated N-cadherin and downregulated E-cadherin. Theophylline treatment reversed these changes induced by BLM, indicating that theophylline suppressed BLM-induced EMT and the accumulation of myofibroblasts (Figure 3D and Appendix A). These experiments demonstrate that theophylline can improve BLM-induced fibrosis by inhibiting profibrotic changes such as lung inflammation and the deposition of extracellular matrix proteins. 

### 2.3. Theophylline Suppresses IL-17 Production in Lung Tissue of BLM-Induced Pulmonary Fibrosis Model

IL-17 promotes the development of pulmonary fibrosis in several animal models. In IPF patients, IL-17 is elevated in the bronchoalveolar lavage and detected in tissues. Thus, we checked the expression and production of IL-17. The protein and mRNA levels of IL-17 were elevated by BLM treatment and significantly reduced by theophylline treatment in lung tissues (Figure 4A,B). These results suggest that the anti-fibrotic effect of theophylline may be related to the suppression of IL-17 production. 

In the early process of fibrosis development, epithelial injury leads to the derangement of alveolar epithelial cells and the accumulation of damaged or dying cells, and then releases danger signals and cytokines, such as IL-1β and IL-6, and leads to the activation of the inflammatory pathway. As the differentiation of Th17 cells from naïve T cells requires TGF-β, IL-6, IL-1β, and IL-23, the mRNA expression of these cytokines was investigated in lung tissues of BLM-induced fibrosis. The expression of TGF-β, IL-6, IL-1β, and IL-23 was increased in the BLM group and significantly decreased in the high dose of theophylline group (Figure 4C and Appendix A). Based on these findings, the expression levels of the transcription factors involved in the Th17 differentiation pathway were further measured in the lung tissues. The mRNA and protein levels of RORγt and IRF4 were increased in the lung tissue of BLM-treated mice but decreased in the theophylline-treated groups (Figure 4D,E and Appendix A). Together, these results indicate that the protective effects of theophylline on BLM-induced lung fibrosis might be at least in part attributed to the inhibition of the Th17 differentiation pathway.

### 2.4. Theophylline Inhibits BLM-Induced IL-6 and IL-1β in BEAS-2B Cells

To determine the possible inhibitory mechanism of theophylline on IL-17 production in lung tissue, we analyzed BLM-induced inflammatory markers in epithelial cells with or without theophylline in vitro. BLM causes an inflammatory response through the production of reactive oxygen species (ROS) as an initial event [15,16]. ROS have been shown to induce NF-kB activation as well as release pro-inflammatory cytokines, including TNF-α, IL-1β, IL-6, and IL-8 [17,18,19]. In this study, the effect of theophylline on the ROS levels was investigated. The level of ROS was significantly elevated after BLM treatment. Theophylline treatment suppressed the levels of ROS in a concentration-dependent manner (Figure 5A). Furthermore, theophylline inhibited the phosphorylation of the NF-κB (p65) induced by BLM (Figure 5B and Appendix A). The expression of IL-6 and IL-1β, Th17 differentiation cytokines, was increased by BLM, but markedly decreased by theophylline (Figure 5C and Appendix A). These results show that theophylline could regulate BLM-induced inflammatory responses and IL-17 production by inhibiting ROS and pro-inflammatory mediators.

### 2.5. Theophylline Inhibits TGF-β Signaling Pathway in TGF-β-Treated BEAS-2B Cells as Well as Lung Tissue of BLM-Induced Pulmonary Fibrosis

In a previous study, we showed an effective experimental condition to alter pro-fibrotic phenotypes by TGF-β treatment in epithelial cells. To investigate the molecular mechanism of the anti-fibrotic effects of theophylline on pulmonary fibrosis, its effect on the TGF-β signaling pathway was examined. In BEAS-2B human lung epithelial cells, TGF-β treatment increased profibrotic phenotype including ColIα1, ColIII, and N-cadherin. Theophylline treatment decreased the expression levels of ColIα1, ColIII, and N-cadherin induced by TGF-β (Figure 6A). The treatment of theophylline also inhibited the mRNA and protein levels of TGF-β-induced IL-6 (Figure 6B). Furthermore, the phosphorylation of Smad2/3 and AKT was significantly increased by TGF-β but reduced by theophylline (Figure 6C and Appendix A). Next, we examined whether theophylline has anti-fibrotic effects in vivo via inhibiting the TGF-β signaling pathway. The phosphorylation of Smad2/3 and AKT was alleviated in the lung lysates of the theophylline-treated mice compared to the BLM-treated mice (Figure 6D and Appendix A). Collectively, these data suggest that theophylline has an anti-fibrotic effect on BLM-induced pulmonary fibrosis by inhibiting the TGF-β/Akt/Smad2/3 pathway.

## 3. Discussion

IPF is a chronic and progressive fetal interstitial lung disease [20]. Since there is no effective drug for the treatment of IPF. IPF patients have a median survival of 2 to 3 years from the time of diagnosis [1,21]. Despite many drugs having gone through clinical trials, only two drugs, pirfenidone and nintedanib, have been approved for IPF treatment, and these do not completely cure the disease. Thus, the development of more effective new drugs is increasingly required to reduce or even halt the progression of the disease and improve the quality of life for IPF patients [22]. Research on new therapeutic targets and treatment strategies for IPF is rapidly increasing, and several candidates for the treatment of IPF are ongoing in preclinical studies and clinical trials [23,24,25,26,27]. In order to develop new treatments in the future, it is essential to elucidate the etiopathogenesis of IPF and develop effective biomarkers for the disease. To find an effective drug for IPF, we chose a drug repositioning strategy. The advantages of this strategy are the lower risk of failure, a reduction in development time, and saving associated costs [28]. 

Theophylline has been already used to treat asthma as a bronchodilator for more than 80 years [29]. However, the use of theophylline for asthma treatment has been limited by side effects and the need for plasma monitoring. The most frequent side effects are headache, nausea and vomiting, increased acid secretion, and gastroesophageal reflux due to phosphodiestease (PDE) inhibition. At high concentrations, the serious side effects of cardiac arrhythmias and seizures occur, presumably due to adenosine receptor antagonism. Nevertheless, in patients with severe asthma, theophylline still remains a very useful add-on therapy. Theophylline is also prescribed at lower concentrations based on its anti-inflammatory effects in COPD [29,30,31]. Recently, theophylline was a possibility in the treatment of COVID-19 patients as it is characterized by bronchodilatory, immunomodulatory, and potentially antiviral mechanisms [32]. Moreover, other xanthine derivatives, such as doxophylline and theobromine, work as a β2-adrenergic receptor (β_2_-AR) agonist, so they derivate airway smooth muscle relaxation via increasing the cAMP and PKA signaling pathways [33]. Much research has reported that β_2_-AR agonists have the effect of anti-inflammation and regulate the symptoms of asthma as a bronchodilator drug [34,35]. In clinical studies, β_2_-AR agonists also improve lung function and increase the ciliary beat frequency in chronic bronchitis [36]. In addition, it has shown that theophylline has anti-inflammatory effects by inhibiting NF-κB activation and enhancing HDAC2 expression in epithelial cells, macrophages, monocytes, and skeletal muscle in animals and human patients with COPD [37,38,39]. In addition, theophylline inhibited TGF-β and collagen production in fibroblast and liver injury animal models [40]. Therefore, we decided to examine the effects of theophylline on pulmonary fibrosis. 

In the present study, we demonstrated that theophylline has anti-fibrotic effects in a BLM-induced fibrosis mouse model. The oral administration of theophylline improved BLM-induced lung inflammatory responses, fibrotic changes, and pulmonary dysfunction. Theophylline treatment also reduced the production of IL-17 and the expression of transcription factors, such as RORγt and IRF4, which were induced by BLM. We further demonstrated that theophylline inhibited collagen deposition and EMT through the suppression of the TGFβ signaling pathway in epithelial cells. Therefore, our results indicate that the anti-fibrotic effects of theophylline are mediated by the regulation of the Th17 differentiation pathway and the TGF-β downstream signaling pathway (Figure 7).

A variety of studies have reported the importance of Th17 cells and IL-17 in IPF. IL-17 was detected in the sera of BLM-treated mice as well as patients with IPF, and lung fibrosis was inhibited by IL-17A antibody treatment [12,41,42,43]. TGF-β and the inflammatory cytokines, IL-6, IL-1β, and IL-23, are required to induce the production of IL-17 and RORγt [44,45,46]. We further analyzed the mRNA expression of Th17 differentiation-related cytokines in lung tissues. In addition to TGF-β, the mRNA expression levels of IL-6, IL-1β, and IL-23 were decreased by theophylline administration in mice. In particular, the level of IL-6 was significantly inhibited compared to other cytokines. Thus, theophylline could inhibit Th17 differentiation via the regulation of Th17 generation factors. 

ROS play a potential role in the pathogenesis of various lung diseases [47,48]. Several studies demonstrated that a marked increase in oxidative DNA damage and ROS-mediated production of oxidized lipids and proteins has been observed in pulmonary fibrosis induced by BLM [49,50]. BLM-induced lung injury generates ROS from alveolar epithelial cells and inflammatory cells [15,16]. ROS promote the NF-κB signaling pathway leading to an inflammatory response [17,51]. NF-κB triggers the expression of various pro-inflammatory genes including IL-1β, IL-6, TNF-α, and IL-8 [17,18,19]. Furthermore, ROS can degrade ECM components, causing ECM remodeling. ROS and ROS-induced ECM fragments also increase the release of TGF-β, thus enhancing fibrogenic processes [52]. In BLM-treated epithelial cells, theophylline treatment inhibited BLM-induced ROS production and NF-κB phosphorylation. In particular, theophylline decreased the expression of Th17 generation factors, IL-1β and IL-6. Therefore, the inhibitory effect of theophylline on Th17 differentiation might be contributed by reducing the BLM-induced release of ROS and pro-inflammatory cytokines in lung epithelial cells.

One of the hypotheses for the process of IPF pathogenesis is aberrant wound healing in response to repeated micro-injuries to the alveolar epithelium as well as an increase in pro-fibrotic cytokines, in particular TGF-β, by inflammatory and epithelial cells [2,21]. TGF-β is the master switch in the fibrotic responses and alteration of the fibrogenic phenotype in fibroblasts, epithelial cells, and immune cells [6,53,54,55,56,57]. In the development of IPF, EMT via the activation of TGF-β stimulates the production of pro-fibrotic mediators, leading to fibroblast differentiation into myofibroblasts, which promotes excessive collagen and ECM deposition [6,58,59,60,61]. Consequently, TGF-β-induced EMT contributes to the development of IPF as the central activator. We previously showed that kurarinone has an anti-fibrotic effect using our experimental system based on the relationship between TGF-β and epithelial cells [62]. Using our experimental condition, theophylline decreased TGF-β-induced pro-fibrotic phenotypes in epithelial cells. Furthermore, IL-6 is a critical cytokine for the differentiation of Th17 cells via the activation of STAT3 and RORγt [63]. Theophylline suppressed TGF-β-induced IL-6 mRNA expression and protein secretion in the epithelial cells. Moreover, TGF-β-induced EMT is mainly mediated by the Smad or non-Smad signaling pathway [6,21,64]. Our results showed that theophylline treatment reduced Smad2/3 and AKT phosphorylation both in vitro and in vivo. Thus, our data indicated that the anti-fibrotic effect of theophylline might be mediated by inhibiting the TGF-β-signaling pathway in lung epithelial cells.

In conclusion, we showed that theophylline attenuated pulmonary fibrosis by inhibiting TGF-β-downstream signaling and Th17 differentiation. Our results suggest that theophylline, which has been already used as an anti-asthmatic drug, may have the potential to be repurposed as a drug for IPF by targeting multiple signaling pathways. Further studies are needed to verify the therapeutic effect of theophylline in fibrosis, and future studies to evaluate its potential as a therapeutic agent in patients with IPF should be encouraged.

## 4. Materials and Methods

### 4.1. Reagents

Bleomycin (BLM, B1141000), used for the in vivo experiment, was purchased from Sigma-Aldrich-Merck (St. Louis, MO, USA). Theophylline (HY-B0809) and PFD (HY-B0673) were obtained from MedChem Express (Monmouth Junction, NJ, USA).

### 4.2. Mice

Eight- to ten-week-old male C57BL6 mice (body weights were 22~24 g) were purchased from DBL (Eumseing, Republic of Korea). All experimental procedures were approved by the Institutional Animal Care and Use Committee of the Korea Research Institute of Bioscience and Biotechnology (Approval number: KRIBB-AEC-20324).

### 4.3. Cell Culture

Human BEAS-2B lung epithelial cells were obtained from the American Type Culture Collection (ATCC, Manassas, VA, USA). The cells were cultured at 37 °C in a 5% CO_2_-humidified incubator and maintained in high-glucose Dulbecco’s modified Eagle medium (DMEM) containing 10% (*v/v*) heat-inactivated fetal bovine serum (FBS), 1 mM sodium pyruvate, 2 mM glutamine, 100 U/mL penicillin, and 50 μg/mL streptomycin. Cells at a confluence of 60–80% were stimulated with the experimental reagents in a medium containing 0.1% FBS as indicated in each experiment. All experiments were replicated three times.

### 4.4. Cell Viability Assay

The cell viability was analyzed using Ez-cytoX (DoGenBio, Seoul, Republic of Korea) according to the manufacturer’s protocol, and the analysis was replicated three times.

### 4.5. Murine Bleomycin-Induced Pulmonary Fibrosis Model

Nine-week-old C57BL6 mice (body weights were 22~24 g) were randomly divided into five groups (n = 6 in each group): normal control group (NC), bleomycin-treated group (BLM), theophylline (20 mg/kg) + BLM-treated group (T20), theophylline (40 mg/kg) + BLM-treated group (T40), and the pirfenidone (150 mg/kg) + BLM-treated group (PFD). The human effective dose (HED) of theophylline (20 and 40 mg/kg) and pirfenidone (150 mg/kg) is 1.6 mg/kg, 3.2 mg/kg, and 12 mg/kg in humans, respectively. At day 0, the mice were anesthetized using a mixture of ketamine and xylazine by i.p. injection. In total, 50 μL of BLM (0.5 mg/kg body weight) was administrated by intratracheal instillation. Sterile saline was administered for the control group instead of the BLM. Theophylline solution and pirfenidone solution were prepared by dissolving them in PBS. The theophylline solution and pirfenidone solution were administered orally five times a week from day 1 to day 13 after BLM administration [40,65,66]. The NC groups received the vehicle only. On day 14, the mice were sacrificed, and the lung tissues were collected for further analysis after a pulmonary function test. The animal experiments were replicated twice. 

### 4.6. Pulmonary Mechanical Function Test

The lung mechanics were assessed using a flexiVent system (SCIREQ, Inc., Montreal, QC, Canada) according to the manufacturer’s protocol. Briefly, the mice were anesthetized using pentobarbital sodium (ENTOBAR^®^, Hanlim Pharm. Co., Ltd., Seoul, Republic of Korea) to suppress spontaneous breathing. A tracheotomy was performed to insert an 18 G cannula into the trachea for connecting the mice with the flexiVent system. The connected mice were ventilated at a respiratory rate of 150 breaths/min and a tidal volume of 10 mL/kg against a positive end-expiratory pressure (PEEP) of 3 cmH_2_O with a computer-controlled small-animal ventilator. The measurement of the respiratory system mechanics was evaluated assuming four different models. Pressure-derived PV curves were generated to determine the distensibility of the entire respiratory system. The total respiratory system resistance (Rrs), compliance (Crs), and elastance of the respiratory system (Ers) were measured by the Snapshot-150. G and H, reflecting the damping and elastance of the lung tissue, respectively, were obtained from Quick Prime-3. All measurements were conducted in mice with closed-chest walls. All data were analyzed using SCIREQ flexiWare (version7.6, service pack 5, SCIREQ)

### 4.7. Bronchoalveolar Lavage Fluid (BALF) and Cell Analysis

BALF was obtained by injecting cold PBS using a tracheal cannula with a 20 G blunt needle. The total number of inflammatory cells was counted by using a hematocytometer after trypan blue staining. To identify the differential cell counts, 100 μL of BALF was centrifuged onto slides using Cytospin (Hanil Science Industrial, Seoul, Republic of Korea). The slides were dried, fixed, and stained using the Diff-Quick staining reagent (B4132-1A; IMEB Inc., Deerfield, IL, USA) [67].

### 4.8. Staining for Histopathological Analysis

After sacrificing the mice on day 14, the left lung tissues were fixed in 10% formalin, embedded in paraffin, and cut into 4 μm-thick sections. The sections were stained with hematoxylin and eosin (Sigma-Aldrich-Merck) or Masson’s trichrome (Abcam, Cambridge, UK).

### 4.9. Hydroxyproline Assay

The hydroxyproline content of the lung tissues was determined by a Hydroxyproline Colorimetric assay kit (BioVision, Milpitas, CA, USA) according to the manufacturer’s instructions. In brief, lung tissues homogenized in dH_2_O were hydrolyzed with concentrated HCl at 120 °C for 3 h, and the supernatants were collected after centrifuging the hydrolyzed homogenates at 10,000× *g* for 3 min. In total, 10 μL of the supernatants were loaded onto a 96-well plate followed by a drying step, and the samples were assessed for hydroxyproline at 560 nm. The data were expressed as micrograms of hydroxyproline per gram of wet lung weight (μg/g wet tissue).

### 4.10. Quantitative Real-Time Reverse Transcription -PCR

Total RNA was extracted from the cell pellets or homogenized lung tissues. For RNA isolation, the TRIzol reagent (Ambion^®^, Thermo Fisher Scientific, Waltham, MA, USA) was used according to the manufacturer’s protocol. The RNA samples were treated with RNase-Free DNase I (Promega, Madison, WI, USA) to avoid DNA contamination in the RNA extraction. The cDNA was synthesized from 1 μg of total RNA (A260/280 ratio > 1.8) using the ReverTra Ace First Strand cDNA Synthesis Kit (Toyobo, Osaka, Japan). For real-time PCR, 1 μL of 20 μL cDNA was used, and the iQ SYBR Green supermix (Bio-Rad, Hercules, CA, USA) and an S1000™ Thermal Cycler (Bio-Rad) were used. Primer sequences are detailed in Appendix A. The real-time PCR was performed for 40 cycles with an annealing temperature of 60 °C. The relative gene expression levels were evaluated by their ratio to GAPDH mRNA or β-actin mRNA [67].

### 4.11. Western Blot Analysis

Cell lysates and tissue lysates were harvested with RIPA buffer (Biosesang, Seongnam, Republic of Korea) containing a protease inhibitor cocktail and a phosphatase inhibitor. The proteins were fractionated on 10% SDS-polyacrylamide gels. The gels were transferred to polyvinylidene fluoride membranes, and the membranes were incubated for 1 h in 5% skim milk in TBS-T buffer. Then, the membranes were incubated with primary antibodies recognizing pAkt (cat#2965), pSmad2/3 (cat#8828), ColIα1 (cat#84336) (Cell Signaling Technology, Danvers, MA, USA), Akt (cat#610861), Smad2/3 (cat#610843) (BD Biosciences, CA, USA), and β-actin (cat#664802, BioLegend, San Diego, CA, USA), followed by incubation with an HRP-conjugated secondary antibody (Jackson ImmunoResearch Laboratories, West Grove, PA, USA). Signals were developed using an enhanced chemiluminescence system (Thermo Fisher Scientific). The optical densities of the target protein bands were analyzed with ImageJ 1.52a (National Institutes of Health, Bethesda, MD, USA).

### 4.12. Measurement of Cytokine Secretion by ELISA

The levels of IL-17 in the lung lysates were analyzed with an IL-17A Mouse Uncoated ELISA Kit (Invitrogen^TM^, Thermo Fisher Scientific) following the manufacturer’s instructions. The levels of IL-6 in the cell supernatants were analyzed with a Human IL-6 ELISA set (BD Biosciences, San Jose, CA, USA) following the manufacturer’s instructions.

### 4.13. ROS Detection Assay

For total intracellular ROS, BEAS-2B cells treated with BLM (3 μg/mL, 2 h) with or without theophylline were stained for 30 min at 37 °C with 10 µM DCFDA then washed and analyzed by a fluorescence microplate reader with excitation/emission at 485 nm/535 nm. 

### 4.14. Statistical Analysis

The data were analyzed and graphed with GraphPad Prism software (ver. 6.07, GraphPad Software, Inc., San Diego, CA, USA) and are presented as means ± SEMs. Statistical analysis was checked by the Kolmogorov–Smirnov test and calculated using analysis of variance (ANOVA) followed by a multiple comparison test with Turkey’s post hoc test. P-values of less than 0.05 were considered statistically significant.

## Figures and Tables

**Figure 1 ijms-24-01019-f001:**
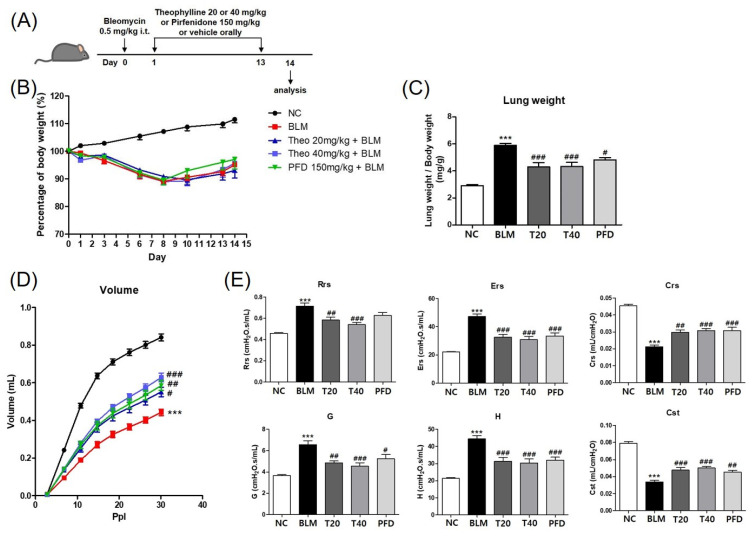
Theophylline improves the mechanical function of the lung in the bleomycin (BLM)-induced pulmonary fibrosis model. (**A**) A schematic illustration of BLM and drug treatment in this study. (**B**) Relative body weight changes in BLM-challenged mice with or without treatment of theophylline and pirfenidone, as indicated. Mice were randomized into weight-matched groups. The relative body weight was calculated as a percentage of that measured on day 0 which is defined as 100%. (**C**) Relative left lung weight in mice challenged with BLM, with or without treatment with theophylline and pirfenidone (PFD). Relative left lung weight was calculated as the ratio of left lung weight (mg) to body weight (g) of each mouse. (**D**) Pressure-volume curves show the correlation of the volume of the respiratory system to incrementing pressure. (**E**) The values of Rrs (resistance of total respiratory system), Ers (elastance of total respiratory system), Crs (compliance of total respiratory system), G (damping of tissue), H (elastance of tissue), and Cst (distensibility of the respiratory system) were shown. NC; normal control mice treated with saline only, BLM; BLM-treated mice, T20 and T40; theophylline (20 and 40 mg/kg) + BLM-treated mice, PFD; pirfenidone (150 mg/kg) + BLM-treated mice. The experiments were conducted two times, with a total of six animals per group. All data are represented as the mean ± SEM (n = 6). *** *p* < 0.001, compared with normal control (NC); ^#^
*p* < 0.05, ^##^
*p* < 0.01, ^###^
*p* < 0.001, compared with BLM group.

**Figure 2 ijms-24-01019-f002:**
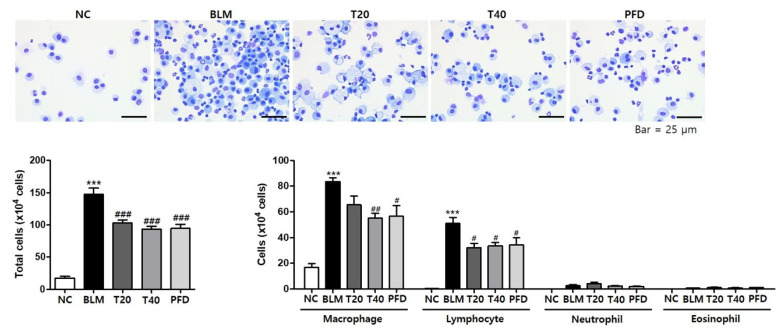
Theophylline suppresses bleomycin (BLM)-induced immune cell accumulation in BALF. Cells in BALF were stained and counted. For differential cell counts, cells were identified as macrophages, lymphocytes, neutrophils, and eosinophils using their cellular and nuclear morphologies. NC; normal control mice treated with saline only, BLM; BLM-treated mice, T20 and T40; theophylline (20 and 40 mg/kg) + BLM-treated mice, PFD; pirfenidone (150 mg/kg) + BLM-treated mice. The experiments were conducted two times, with a total of six animals per group. All data are represented as the mean ± SEM (n = 6). *** *p* < 0.001, compared with normal control (NC); ^#^
*p* < 0.05, ^##^
*p* < 0.01, ^###^
*p* <0.001, compared with the BLM group.

**Figure 3 ijms-24-01019-f003:**
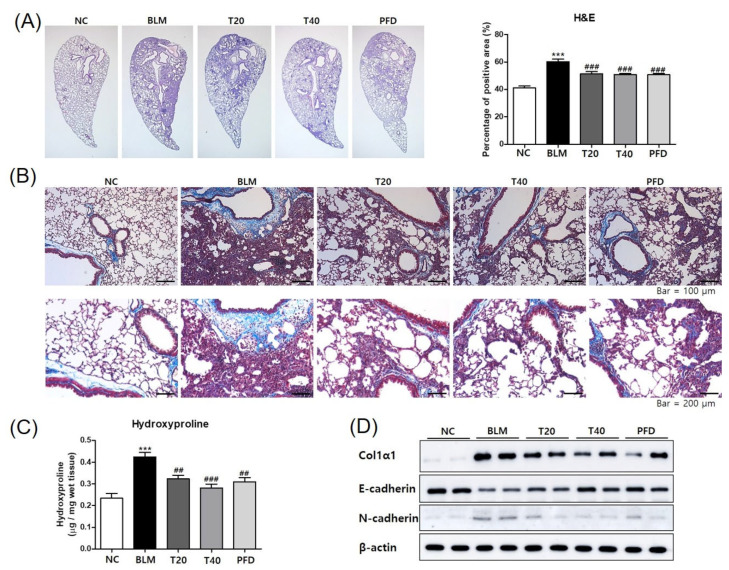
Theophylline attenuates the fibrotic changes in lung tissue of bleomycin (BLM)-induced lung fibrosis mice model. (**A**) Lung tissues were fixed, sectioned at 4 μm, and stained with H&E. Densities of an H&E image were quantitated using the ImageJ tool. (**B**) Histological analysis in the lungs was performed by Masson’s trichrome stain with collagen staining blue. (**C**) The amounts of hydroxyproline in the lung tissues were analyzed as quantitation of collagen. (**D**) The protein levels of ColIα1, E-cadherin, and N-cadherin in the lung tissues were analyzed by Western blot. NC; normal control mice treated with saline only, BLM; BLM-treated mice, T20 and T40; theophylline (20 and 40 mg/kg) + BLM-treated mice, PFD; pirfenidone (150 mg/kg) + BLM-treated mice. The experiments were conducted two times, with a total of six animals per group. All data are represented as the mean ± SEM (n = 6). *** *p* < 0.001, compared with normal control (NC); ^##^
*p* < 0.01, ^###^
*p* < 0.001, compared with BLM group.

**Figure 4 ijms-24-01019-f004:**
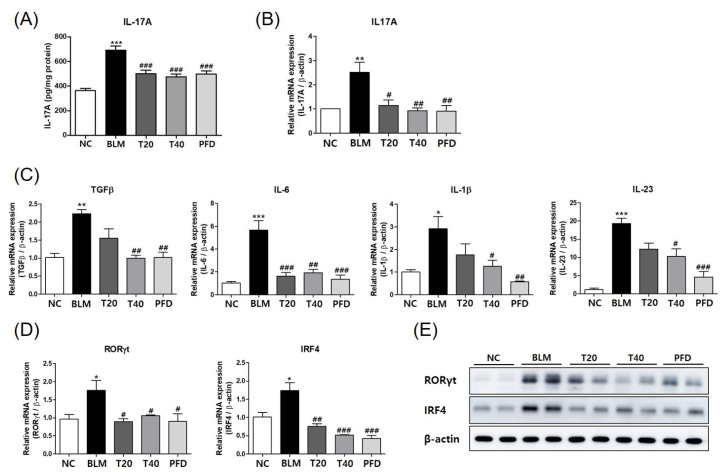
Theophylline suppresses the IL-17 production in the lung tissue of the bleomycin (BLM)-induced pulmonary fibrosis model. (**A**) Levels of IL-17A in lung tissues sampled from mice of 5 study groups detected by ELISA. (**B**) The mRNA levels of IL-17A in lung tissues were determined by real-time RT-PCR. (**C**) The mRNA levels of TGF-β, IL-6, IL-1β, and IL-23 in the lung tissues were determined by real-time RT-PCR. (**D**) The mRNA levels of RORγt and IRF4 in the lung tissues were determined by real-time RT-PCR. The mRNA expression data were normalized to β-actin mRNA expression. (**E**) Protein levels of RORγt and IRF4 in the lung tissue were analyzed by Western blot. Representative results from independent experiments were presented. NC; normal control mice treated with saline only, BLM; BLM-treated mice, T20 and T40: theophylline (20 and 40 mg/kg) + BLM-treated mice, PFD; pirfenidone (150 mg/kg) + BLM-treated mice. The experiments were conducted two times, with a total of six animals per group. All data are represented as the mean ± SEM (n = 6). * *p* < 0.05, ** *p* < 0.01, *** *p* < 0.001, compared with normal control (NC); ^#^ *p* < 0.05, ^##^ *p* < 0.01, ^###^ *p* < 0.001, compared with BLM group.

**Figure 5 ijms-24-01019-f005:**
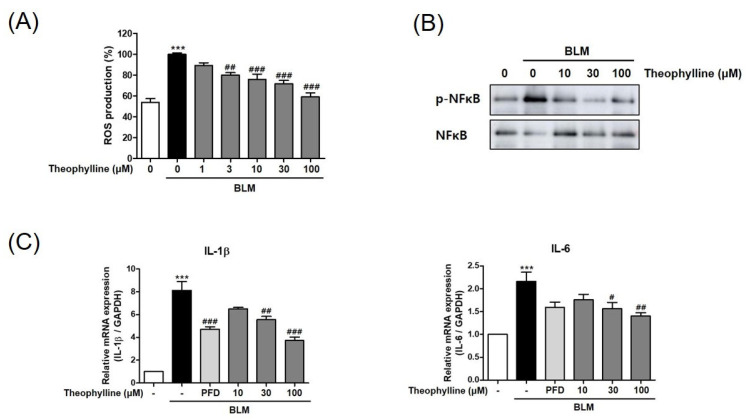
Theophylline inhibits the bleomycin (BLM)-induced IL-6 and IL-1β in BEAS-2B cells. (**A**) The ROS levels in BEAS-2B cells treated with BLM (3 μg/mL, 2 h) with or without theophylline were measured using DCFDA. (**B**) Phosphorylation of NFκB in BEAS-2B cells treated with BLM (3 μg/mL, 1 h) with or without theophylline was analyzed by Western blot. (**C**) The mRNA and protein levels of IL-1β and IL-6 in BEAS-2B cells treated with BLM (3 μg/mL, 24 h) with or without theophylline were determined by real-time RT-PCR. The mRNA expression data were normalized to GAPDH mRNA expression. All experiments were repeated three times and representative results from three independent experiments were presented. Statistics represented as mean ± SEM of each group. *** *p* < 0.001, compared with untreated control; ^#^ *p* < 0.05, ^##^ *p* < 0.01, ^###^ *p* < 0.001, compared with the BLM-treated group.

**Figure 6 ijms-24-01019-f006:**
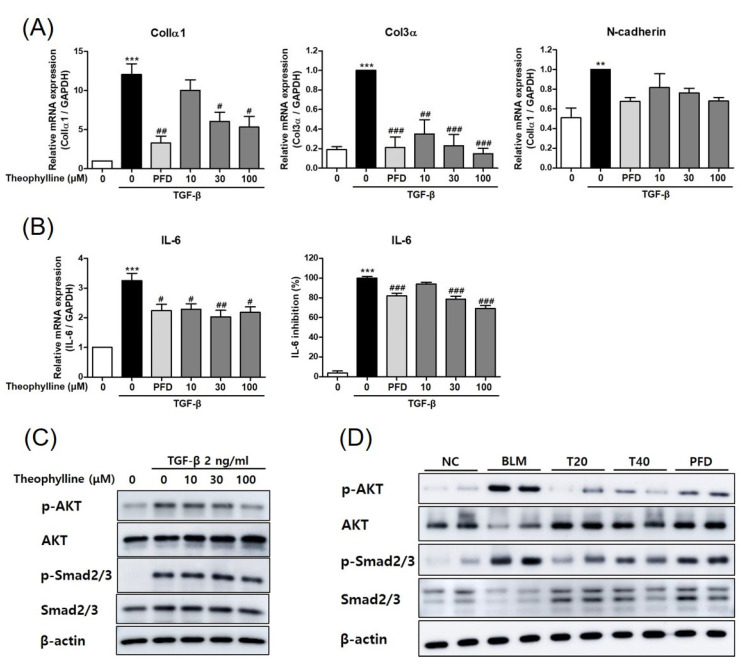
Theophylline inhibits the TGF-β signaling pathway in TGF-β-treated BEAS-2B cells and the lung tissue of bleomycin (BLM)-induced pulmonary fibrosis. (**A**) The mRNA levels of ColIα1, ColIIIα, and N-cadherin in BEAS-2B cells treated with TGF-β (2 ng/mL, 24 h) with or without theophylline were measured by real-time RT-PCR. (**B**) The mRNA and protein levels of IL-6 in BEAS-2B cells treated with TGF-β (2 ng/mL for 24 h) with or without theophylline were determined by real-time RT-PCR and ELISA. The mRNA expression data were normalized to GAPDH mRNA expression. (**C**) Phosphorylation of Smad2/3 and AKT in BEAS-2B cells treated with TGF-β (2 ng/mL for 1 h) with or without theophylline were analyzed by Western blot. (**D**) Protein levels of phosphorylated Smad2/3 and AKT in lung tissue were analyzed by Western blot. NC; normal control mice treated with saline only, BLM; BLM-treated mice, T20 and T40: theophylline (20 and 40 mg/kg) + BLM-treated mice, PFD; pirfenidone (150 mg/kg) + BLM-treated mice. The animal experiment was conducted two times, with a total of six animals per group. The cell experiments were repeated three times and representative results from three independent experiments were presented. Statistics represented as mean ± SEM of each group. ** *p* < 0.01, *** *p* < 0.001, compared with untreated control; ^#^ *p* < 0.05, ^##^ *p* < 0.01, ^###^ *p* < 0.001, compared with TGFβ-treated group.

**Figure 7 ijms-24-01019-f007:**
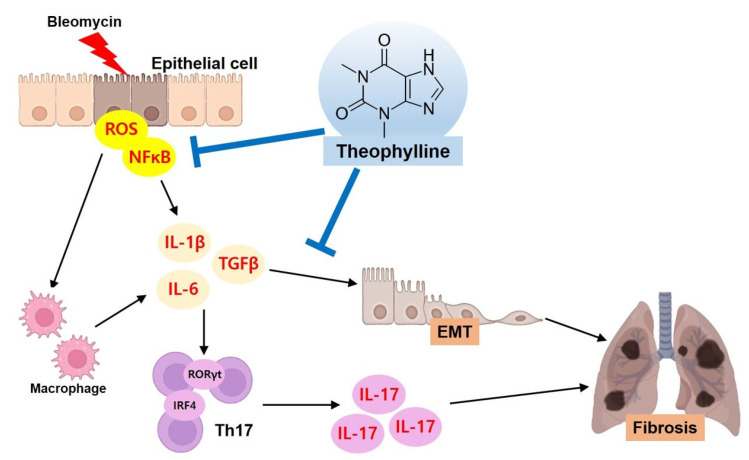
Possible mechanism of the theophylline in a bleomycin (BLM)-induced fibrosis model. BLM-stimulated epithelial cells induce ROS production and NF-κB activation. In response to ROS and the NF-κB signaling pathway, epithelial cells secrete pro-inflammatory and pro-fibrotic cytokines, TGF-β, IL-6, and IL-1β. These cytokines promote the differentiation of Th17 cells as well as EMT. Secreted IL-17 and increased EMT exacerbate pulmonary inflammation and fibrosis. The effects of theophylline are represented by the blue arrows. Theophylline inhibits IL-17 production, and TGF-β-induces pro-fibrotic changes.

## Data Availability

Data are contained within the article or Appendix A.

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
