# Peer review of "Theophylline Attenuates BLM-Induced Pulmonary Fibrosis by Inhibiting Th17 Differentiation"

_ijms, 2023, doi:10.3390/ijms24021019_

Round 1

Reviewer 1 Report

Hello, 

Thank you for submitting the manuscript titled “Theophylline attenuates BLM-induced pulmonary fibrosis by inhibiting Th17 differentiation”. The manuscript is written well overall. There are few points that authors should consider before next submission.

Minor:

1.       Manuscript should elaborate recent advancement of the treatment in IPF and where does the Theophylline stand in terms of the treatment for other respiratory diseases and its limitation.

2.       As authors have said, theophylline has been in the treatment regime for long time though not the first line therapy. What theophylline and other xanthine derivatives such as doxophylline has to offer to improve the lung status. Please discuss.

3.       Why did the author only consider using young mice. The experiments should have been done with old mice such as 22-24 weeks to see the IPF changes with theophylline.

4.       Please correct the manuscript by removing the proof editing comments on page 6, 7 and 8.

Reviewer 2 Report

The manuscript entitled “Theophylline attenuates BLM-induced pulmonary fibrosis by inhibiting Th17 differentiation” addresses the beneficial effects of theophylline, a classical therapy for asthma, against bleomycin-evoked pulmonary fibrosis and associated molecular mechanisms. Initially, the authors proved that theophylline improved mechanical pulmonary functions and attenuated pulmonary histopathological aberrations in mice. Then, the authors proceeded to some implicated mechanisms. To this end, the authors demonstrated that suppression of IL-17 production, ROS generation, nuclear factor kappa B activation, and Smad2/3 and AKT phosphorylation were implicated in the observed favorable attenuation of pulmonary fibrosis.

The manuscript is clearly written, and the current findings are interesting.

Comments:     

1) In section 4.5. (Murine bleomycin-induced pulmonary fibrosis model), how did the authors decide on the dose of bleomycin in mice? Authors are advised to address this point and add the answers/proper citations in section 4.5.

2) How did the authors decide on the dose of theophylline (20 and 40 mg/kg) in mice? How is the dose of theophylline relevant to the human dose using the Human effective dose (HED) formula= animal dose x animal Km/ human Km (Nair AB, Jacob S. A simple practice guide for dose conversion between animals and humans. J Basic Clin Pharm. 2016 Mar;7(2):27-31). Authors are advised to address this point and add the answers/proper citations in section 4.5.

3) How did the authors decide on the dose of pirfenidone (150 mg/kg) in mice? How is the dose relevant to the human dose using the Human effective dose (HED) formula= animal dose x animal Km/ human Km (Nair AB, Jacob S. A simple practice guide for dose conversion between animals and humans. J Basic Clin Pharm. 2016 Mar;7(2):27-31). Authors are advised to address this point and add the answers/proper citations in section 4.5.

4) To avoid confusion of readers, the authors are advised to clearly describe in section 4.5 the timeline for administration of pirfenidone. What was the used route? Authors are advised to address this point and add the answers to the comment in section 4.5.

5) In the experimental design, why did not the authors incorporate an additional gp (NC + theophylline)? This group may reveal any potential toxicity of the tested dose of theophylline. This may seem important in light of the known tight therapeutic index of theophylline and its potential toxicity.

6) The authors are advised to add the cat no. for the used chemicals and antibodies.

7) In Figure 6A (gene expression data), have the authors considered that the gene expression assays of Collα1, Col3α, and N-cadherin using RT-PCR may not be adequate for quantifying the target signals? In fact, the mRNA expression may not necessarily reflect the corresponding protein levels due to the post-translational modifications. Detecting the protein signals using ELISA or Western blotting is expected to give more reliable data than gene expression assays.

8) Likewise, in figure 5C, the gene expression data for IL-1beta and IL-6 may not be adequate for quantifying the target signals.

9) the above comments are also applicable to Fig. 4C.

10) In figure legends, qRT-PCR is missing biological (how many samples were used per experimental group) and technical repeat information (whether each sample was repeated during the assay). Please, add these data to the relevant figure legends.

11) In qPCR, did the authors check the RNA quality with A260/280, and perform an RT negative control to ensure no DNA contamination in the RNA extraction? Please, add these data in the material and methods section.

12) The author should mention the amount of cDNA used for qRT-PCR. Please, add these data in the material and methods section.

13) The author should mention the qRT-PCR condition such as annealing temp. and the number of cycles. Please, add these data in the material and methods section.

14) In qPCR: The authors are advised to add a table (even as supplementary material) to describe the primer sequence, gene accession number, and amplicon size for all target genes. Please, add these data to the material and methods section

15) In the statistical analysis section, did the authors check data normality and homogeneity before proceeding to one-way ANOVA? Authors are advised to address this point and add the answers to the comment in the material and methods section.

16) To make all figure legends stand-alone, authors are advised to add the full name of all the used abbreviations at the end of each legend.

17) In figure legends, the authors are advised to add the number of animals/replicates from which data were extracted. Authors are advised to address this point and add the answers to the comment to all the relevant figure legends.

18) In figure legends, the authors are advised to describe the number of replicates used in Western blotting. Moreover, were the data extracted from independent samples?

19) To avoid readers’ confusion; the authors are advised to provide higher-magnification pics in figures 3A and B. Please, add

20) In the discussion section, authors are advised to describe the reported adverse effects of theophylline from the literature.

21) More recent 2022 references are advised to be added to the current manuscript.  

Reviewer 3 Report

In this article, Soo-Jin Park et al. investigated the effect of theophylline using the bleomycin model of lung fibrosis in mice. To cut a long story short, the authors only demonstrated an anti-inflammatory effect of theophylline in this model as the animals were treated from day 1 after the bleomycin challenge (during the inflammatory phase…). It is well acknowledged in the field, for more than a decade, that only treatment under a therapeutic protocol (from day 7 at least) is relevant regarding the potential anti-fibrotic effect of a given compound (see the seminal paper of Moeller et al in 2008 (14 years ago…PMID 17936056) and the guidelines from the ATS published in 2017 (PMID 28459387)). 

Round 2

Reviewer 3 Report

The authors did not address my initial concerns in the revised version.

They should have, at least, included the experiments that they are currently conducting (according to the response letter) with respect to the effects of theophylline after the inflammatory phase in their model. In addition, they should keep in mind that they would have to confirm their initial results in another model of lung fibrosis to be thoroughly.

Round 3

Reviewer 3 Report

The authors still did not address my initial concerns in this new revised version and as stated by Moeller et al. in 2008 (PMID: 17936056) : "it is critical to distinguish between drugs interfering with the inflammatory and early fibrogenic response from those preventing progression of fibrosis, the latter likely much more meaningful for clinical application".